# Comprehensive Geriatric Care in Older Hospitalized Patients with Depressive Symptoms

**DOI:** 10.3390/geriatrics8020037

**Published:** 2023-03-12

**Authors:** Ulrich Niemöller, Andreas Arnold, Thomas Stein, Martin Juenemann, Mahmoud Farzat, Damir Erkapic, Josef Rosenbauer, Karel Kostev, Marco Meyer, Christian Tanislav

**Affiliations:** 1Department of Geriatrics and Neurology, Diakonie Hospital Jung Stilling Siegen, Germany Wichernstrasse 40, 57074 Siegen, Germany; ulrich.niemoeller@diakonie-sw.de (U.N.);; 2Departement of Neurology, Justus Liebig University, 35392 Giessen, Germany; 3Department of Urology, Diakonie Hospital Jung Stilling Siegen, 35392 Siegen, Germany; 4Department of Cardiology, Diakonie Hospital Jung Stilling Siegen, 35392 Siegen, Germany; 5Epidemiology, Philipps University Marburg, 35037 Marburg, Germany

**Keywords:** depression, comprehensive geriatric care, elderly patients

## Abstract

Background/Objectives: Depressive symptoms (DS) may interfere with comprehensive geriatric care (CGC), the specific multimodal treatment for older patients. In view of this, the aim of the current study was to investigate the extent to which DS occur in older hospitalized patients scheduled for CGC and to analyze the associated factors. Furthermore, we aimed to investigate whether DS are relevant with respect to outcomes after CGC. Methods: For this retrospective study, all patients fulfilling the inclusion criteria were selected by reviewing case files. The main inclusion criterion was the completion of CGC within the defined period (May 2018 and May 2019) in the geriatrics department of the Diakonie Hospital Jung-Stilling Siegen (Germany). The Geriatric Depression Scale was used to asses DS in older adults scheduled for CGC (0–5, no evidence of DS; 6–15 points, DS). Scores for functional assessments (Timed Up and Go test (TuG), Barthel Index, and Tinetti Gait and Balance test) were compared prior to versus after CGC. Factors associated with the presence of DS were studied. Results: Out of the 1263 patients available for inclusion in this study, 1092 were selected for the analysis (median age: 83.1 years (IQR 79.1–87.7 years); 64.1% were female). DS (GDS > 5) were found in 302 patients (27.7%). The proportion of female patients was higher in the subgroup of patients with DS (85.5% versus 76.3%, *p* = 0.024). Lower rates of patients diagnosed with chronic pulmonary obstructive disease were detected in the subgroup of patients without DS (8.0% versus 14.9%, *p* = 0.001). Higher rates of dizziness were observed in patients with DS than in those without (9.9% versus 6.2%, *p* = 0.037). After CGC, TuG scores improved from a median of 4 to 3 (*p* < 0.001) and Barthel Index scores improved from a median of 45 to 55 (*p* < 0.001) after CGC in both patients with and without DS. In patients with DS, the Tinetti score improved from a median of 10 (IQR: 4.75–14.25) prior to CGC to 14 (IQR 8–19) after CGC (*p* < 0.001). In patients without DS, the Tinetti score improved from a median of 12 (IQR: 6–7) prior to CGC to 15 (IQR 2–20) after CGC (*p* < 0.001). Conclusions: DS were detected in 27.7% of the patients selected for CGC. Although patients with DS had a poorer baseline status, we detected no difference in the degree of improvement in both groups, indicating that the performance of CGC is unaffected by the presence of DS prior to the procedure.

## 1. Background

Geriatric departments have been established in order to treat elderly patients specifically. In accordance with standardized protocols, a multimodal treatment for these patients known as comprehensive geriatric care (CGC) follows specific recommendations focusing on medical requirements but also takes into account functional outcomes [1,2,3]. CGC is characterized as the multidisciplinary approach for developing individual treatment strategies within a team, including members from different medical professions such as experienced physicians, occupational therapists, physiotherapists, speech therapists, dietitians, psychologists, and social workers [1,2,3]. There are various factors that may have an impact on medical procedures, determining both their efficacy and outcome [4,5,6]. Among others, depression has been described as a major factor interfering with recovery in patients with different diseases [7,8,9,10,11]. Furthermore, depression is one of the main reasons patients require support in later life, with a prevalence ranging around 20% [12]. Therefore, this factor is relevant and should be considered when planning rehabilitation measures for older individuals. For instance, depressive symptoms were described as being associated with poor recovery in patients after hip fracture [7]. A number of recommendations have been developed in order to positively influence the course of the disease in patients with breast cancer, including psychosocial interventions for handling depressive symptoms [9]. In older adults with cancer, depression was identified as one of the factors to focus on in order to optimize the rehabilitation of the patient to the greatest extent possible [8]. In patients with myocardial infarction, depression is a facilitator for health care consumption after the acute event [11]. By contrast, co-morbid depression is associated with a high risk of failure to make a long-term recovery in stroke patients [13]. On the other hand, CGC seems to have an impact on depressive symptoms. Shu-Fen Su and colleagues demonstrated an amelioration of depressive symptoms in patients undergoing CGC after hip fracture. The study also found that CGC resulted in a lower rate of subsequent emergency department visits, indicating that specific depression management as part of an individualized care plan should be considered when planning comprehensive geriatric care [14]. However, the observed positive effects in different situations related to the utilization of antidepressant drugs in patients with a comorbid depression also urge the necessity not to underestimate this disorder [15,16,17]. In summary, focusing on depressive symptoms in older adults is of particular relevance as specific management of these symptoms as part of comprehensive geriatric care renders better long-term outcomes and also because specific therapeutic measures may help to combat depressive symptoms in affected patients scheduled for CGC.

While it has already been shown that CGC treatment lowers the degree of depressive symptoms, the extent to which preexisting depression influences CGC and its outcome remains a topic of debate [14]. It is for this reason that in the present study we aimed to investigate depressive symptoms in older hospitalized adults scheduled for CGC in a large geriatric unit. Furthermore, we aimed to investigate the factors associated with depressive symptoms and the degree of improvement after CGC in older patients depending on the presence of depressive symptoms.

## 2. Methods

### 2.1. Patients and Study Design

In this retrospective study, all patients fulfilling the inclusion criteria were selected by reviewing case files. The main inclusion criterion was the completion of CGC within the defined period (May 2018 and May 2019) in the geriatrics department of the Diakonie Hospital Jung-Stilling Siegen (Germany); a subset of patients with complete documentation was included in the analyses (Figure 1).

After a detailed assessment, patients were referred for CGC where appropriate. Patients were referred to our department (a 50-bed geriatric department in the German region of South Westphalia) for CGC from the emergency department or from other external or in-house departments and general practitioners. The analysis for identifying factors associated with the presence of depressive symptoms was based on a cross-sectional design; the evaluation of benefits in functionality after the completion of CGC implicated a longitudinal design investigating the cohort of geriatric patients (those with depressive symptoms and those without) treated in the department.

### 2.2. Comprehensive Geriatric Care (CGC)

A standardized geriatric assessment regarding patients’ mobility, ability to cope with daily tasks, cognitive function, and emotional and social condition was performed on hospital admission and also on discharge. CGC was defined as a multicomponent intervention addressing multiple health domains to develop a person-centered therapeutic plan satisfying acute medical requirements and rehabilitation needs. The selected treatment regime was adapted to reflect patients’ deficits and was continuously re-evaluated. CGC included treatment by an interdisciplinary team consisting of geriatric nursing staff, physiotherapists, occupational therapists, speech therapists, and psychologists under the supervision of an experienced geriatrician. A minimum of 20 regular treatment units of physiotherapy and/or occupational therapy were scheduled for each patient. According to the CGC protocol, a minimum period of 2 weeks was allocated to complete the required therapy units. Medical visits were carried out daily by a geriatrician; medications were adapted and diagnostic procedures undertaken if necessary. Team conferences took place weekly on the basis of a standardized protocol to discuss treatment progress. Patients who completed the required number of CGC therapy units were considered for inclusion in the present analysis.

### 2.3. Data Documentation and Assessments

All relevant data pertaining to patients’ care and medical treatment were documented and recorded systematically and used regularly as the basis for interdisciplinary conferences, quality assurance measures, and billing calculations. Baseline demographic parameters and relevant information regarding patients’ morbidity and functional outcome were used for the current analysis. These included: age, sex, medical comorbidities (hypertension, diabetes mellitus, heart insufficiency, renal insufficiency, coronary heart disease, peripheral artery disease, atrial fibrillation, chronic obstructive pulmonary disease, dementia, Parkinson’s disease, previous stroke, current fracture, osteoporosis and vitamin B deficiency), information on short-term adverse events during hospitalization (diffuse pain, delirium, pneumonia, urinary tract infection, dizziness, deep vein thrombosis, pulmonary embolism, electrolyte imbalance, hypokalemia, and hyponatremia), and results of functional assessments on admission and discharge. In particular, data regarding walking ability as assessed using the Timed Up and Go test on admission and discharge were used for the current analysis. Data obtained from commonly used assessments in the geriatric field were also used for the current investigation (Barthel Index, Tinetti Geriatric Assessment, Geriatric Depression Scale (GDS), Mini-Mental-State Examination (MMSE)).

### 2.4. Geriatric Depression Scale

The Geriatric Depression Scale (GDS) is a screening tool used to identify symptoms of depression in older adults, which consists of a 15-item questionnaire appropriate for assessing older healthy individuals as well as older individuals with mild to moderate cognitive impairment. The procedure is performed routinely as part of a standardized protocol in all patients scheduled for CGC. Patients scoring 0–5 points on the scale are considered as showing no evidence of depressive symptoms, those with a score > 5 are deemed as showing evidence of depressive symptoms, and those scoring 6–10 are considered to have moderate to mild depression, while scores of 11–15 indicate severe depression [18].

### 2.5. Assessment of Walking Ability (Timed Up and Go Test, TuG)

The TuG is a simple, widely used test to assess a person’s mobility, focusing on both the dynamic and static balance. The TuG assesses the time a person needs to rise from a chair, walk three meters, turn around, walk back, and sit down again [19]. Based on the time required to complete the test, we categorized the results into 5 classes: (5) no walking ability at all; (4) >30 s needed to complete the test; (3) 20–29 s needed to complete the test; (2) 10–19 s to complete the test, and (1) <10 s needed to complete the test. The TuG assessments considered for our analysis were performed prior to and after CGC.

### 2.6. Assessment of Balance and Gait (Tinetti Balance and Gait Test)

The TBGT is a commonly used tool for assessing balance and gait dysfunction and fall risk in elderly patients. Patient balance is assessed in a sitting and standing position, when rising from and sitting down into a chair, rotating through 360°, and applying slight pressure on patient’s chest. The gait function is evaluated by reporting the length, height, symmetry, and continuity of the steps. Each item corresponds to 0–2 points, and the maximum possible score in the Tinetti test is 28. The lower the Tinetti score, the higher the risk of falling and the greater the chances of mobility restriction [20].

### 2.7. Mini-Mental-State Examination (MMSE)

MMSE is a commonly used tool for assessing cognitive impairment in adult patients. The MMSE is an 11-question assessment used by clinical healthcare professionals in a clinical setting. It covers various cognitive domains: thinking, communication, understanding, and memory. The maximum score for the MMSE is 30. A score of 25 or higher is classed as normal. If the score is below 24, the result is usually considered to be abnormal, indicating possible cognitive impairment [21].

### 2.8. Assessment of Basic Activities of Daily Living (Barthel Index)

The BI is widely used in clinical practice to assess patients’ disability. It includes ten different elements (ingestion, bed/chair transfer, dressing, walking, grooming, climbing stairs, use of toilet, bathing, continence of bowels, controlling bladder) relating to patients’ basic ADL and mobility. The examiner allocates scores of between 0 and 100 for each item according to the patient’s ability; the higher the value, the better the functional status [22,23].

### 2.9. Statistical Analyses

All data for continuous variables were expressed as median and interquartile ranges. Categorical variables were reported as frequencies and percentages. Nonparametric data were analyzed by applying a two-tailed Mann–Whitney U test. Fisher’s exact test was used to compare relative frequencies. When comparing patients with DS to those without, we included categorical variables associated in the univariate analysis in the logistical regression analysis for identifying factors independently associated with the presence of DS. Statistical analyses were performed using the SPSS software (version 22.0, IBM Corporation, Armonk, NY, USA).

### 2.10. Ethical Approval

The study was reviewed by the local ethics committee (Ethics Committee of the Medical Association of Westphalia-Lippe and the Westphalian Wilhelms-University Muenster) and approval was granted for analysis of the data (protocol number: 2019-517-f-S).

## 3. Results

Of the 1263 patients hospitalized in our department, 1099 patients underwent CGC. A complete GDS assessment was documented for 1092 patients (Figure 1). These were included in the current analysis (median age: 83.1 years (IQR 79.1–87.7 years); 64.1% were female). Depressive symptoms (GDS > 5) were found in 302 patients (27.7%) and severe depressive symptoms (GDS 11–15) in 42 patients (3.9%), while 260 patients (23.8%) presented with moderate to mild depressive symptoms (Figure 2 and Table 1).

In patients both with and without depressive symptoms, the TuG score improved from a median value of 4 prior to CGC to 3 after CGC (*p* < 0.001). The Barthel Index also improved in patients with and without depressive symptoms, increasing from a median value of 45 prior to CGC to 55 after CGC (*p* < 0.001). In patients with depressive symptoms, the Tinetti score improved from median 10 (IQR: 4.75–14.25) prior to CGC to 14 (IQR 8–19) after CGC (*p* < 0.001), while patients without depressive symptoms improved from median 12 (IQR: 6–7) prior to CGC to 15 (IQR 2–20) after CGC (*p* < 0.001). Results are summarized in Table 2.

Comparing patients with depressive symptoms to those without, we observed a higher proportion of female patients in the subgroup of patients with depressive symptoms (85.5% versus 76.3%, *p* = 0.024). Lower rates of patients diagnosed with chronic pulmonary obstructive disease were detected in the subgroup of patients without depressive symptoms (8.0% versus 14.9%, *p* = 0.001). Higher rates of dizziness were observed in patients with depressive symptoms than in those without (9.9% versus 6.2%, *p* = 0.037). Walking ability prior to CGC was slightly better in the subgroup of patients with depressive symptoms (TuG mean 4.1 versus TuG mean 3.8, *p* = 0.001). The ability to cope with daily tasks prior to CGC was worse in the subgroup of patients with depressive symptoms than in those without (Tinetti score median 10 versus Tinetti score median 12, *p* = 0.001). Results are summarized in Table 2. To calculate associations in the equation, parameters associated in the univariate analysis were entered in a stepwise logistical regression analysis. Female sex and chronic obstructive pulmonary disease remained independent predictors for depressive symptoms in elderly patients scheduled for CGC. In addition to this calculation and in order to overcome the dichotomization of GDS in the initial analysis, a linear logistic regression analysis with correction for age was performed. The factors female sex, chronic obstructive pulmonary disease, and the diagnosis of dementia explained values detected by GDS in our patients (see also Table 3).

## 4. Discussion

In our study, we identified depressive symptoms which could impact the benefit of comprehensive geriatric care in 27.7% of patients selected for the treatment. However, the majority of patients (72.3%) selected for inclusion did not display depressive symptoms. Improvements in walking ability and basic activities of daily living noted after the CGC were comparable in both patients with and those without depressive symptoms. With regard to balance and gait abilities, patients with depressive symptoms started from a lower baseline but still improved during CGC treatment, ultimately achieving the same functional level as those without depressive symptoms. The presence of depressive symptoms was associated with female sex and chronic obstructive pulmonary disease; dementia and the initial functional status prior to CGC may also be of relevance.

A diagnosis of depression can be expected in more than 20% of older adults, and individuals are considered more vulnerable to developing the condition with increasing negative life events such as chronic health conditions or loss of ability to cope with daily living tasks [24,25,26]. The proportion of depressive disorders is even higher in older patients hospitalized for the treatment of an acute disease, ranging from 20 to 50% [27,28,29,30]. The population investigated is not homogeneous in nature due to the different admission procedures for entry to our department and therefore represents a hospitalized population of older patients with different characteristics and illnesses. As a result, the frequency of 27.7% we determined for patients with depressive symptoms is in line with values indicated previously in the literature [26,27,28,29,30]. However, this finding is of major importance in the context of CGC as a specific focus on the treatment of depressive symptoms as part of the therapy is necessary for this particular proportion of patients. It is well known that depression influences cognitive ability, especially among older adults, and this phenomenon, known as cognitive frailty, needs to be taken into consideration when planning therapeutic procedures such as CGC [24]. Apart from medical measures such as psychotherapeutic interventions and specific drugs, technically supported procedures such as digitally delivered cognitive behavioral therapies could be considered to support CGC [31,32]. Nevertheless, further research is needed in this regard to investigate the benefits of specific interventions in older patients with depression who are scheduled for CGC. Measures in the field of digital mental interventions are particularly promising and should be considered for further evaluation [33]. Experience with digital technologies gained during the COVID-19 pandemic might help to support this approach [34].

In our study, gait and balance and walking ability were worse in patients with depression than in those without when assessed prior to CGC, indicating that depression in older adults might have a direct influence on the ability to cope with daily tasks [35]. A depressive disorder could impact the cognitive ability necessary to perform functional tests, although it should also be noted that depressive symptoms in older adults are generally associated with poorer functional status [35]. Although both aspects might explain the findings depicted in our study, CGC rendered an improvement in abilities of daily living irrespective of the presence of depressive symptoms, as indicated by the TuG, Barthel Index, and Tinetti test. As far as balance and gait ability are concerned, our data indicate that while patients with depressive symptoms may start from a lower baseline than those without depressive symptoms, both groups of patients still achieve comparable levels after the CGC.

In our groups of older adults, we identified female sex as being associated with the presence of depressive symptoms [36]. This could be because females are generally more prone to experiencing depressive disorders, but this finding could also be determined by differences in how women and men deal with disorders [37,38]. We also determined higher rates of patients with depressive symptoms among patients with chronic pulmonary obstructive disease; this is in line with previous investigations which indicate that depression occurs commonly in individuals with this disease [39,40]. As previously described, depressive symptoms in patients with hip fracture may interfere with subsequent recovery [7]. By contrast, in our study, patients with fractures were distributed equally among patients with depressive symptoms (27.5%) and those without (27.3%). Our results therefore suggest that a recent fracture which initially led to hospitalization was not the crucial determinant for the development of depressive symptoms, indicating that the differences depicted are not due to the presence of a fracture. Patients with depressive symptoms were more likely to experience dizziness, which is probably due to the mutual relationship between both conditions, already well described in the literature [41].

The major strengths of the present investigation are the large number of participants who received CGC in accordance with standardized protocols and the detailed documentation of all relevant parameters as part of the clinical process. However, there are also a number of limitations that must be taken into account. The major limitation of this study is the fact that no control group with regular subject-specific treatment was available; a control group including patients not undergoing CGC could help to better estimate the effects (improvements in functionality) of the procedure in patients with depressive symptoms as well as those without. On the other hand, it is difficult to design such a control group in the knowledge that to do so would mean withholding comprehensive geriatric care from those patients in whom the procedure was previously considered beneficial. In addition, parameters regarding the quality of life were not routinely documented and no information on this aspect was therefore available for study. The possibility of a selection bias should also be mentioned because those patients likely to benefit most from CGC were selected for the treatment as part of the geriatric pre-assessment. Since our study was conducted using clinical routine data, there is a lack of rigorous settings for proving therapy effects are missing. No causal relationships could therefore be reported due to the retrospective nature of the analysis; instead, we focused solely on associations.

Our results indicate that CGC is of great benefit to elderly patients, improving all aspects of functionality and unveiling the potential determining factors for a favorable outcome. Although our data are derived from clinical settings, and therefore more prone to differential bias, they elucidate the benefits of CGC under real-world conditions. From the clinician’s perspective, however, selection bias in choosing candidates for the procedure remains inevitable.

## 5. Conclusions

Depressive symptoms were detected in almost one-third of the patients selected for comprehensive geriatric care. As this finding might interfere with the CGC procedure, specific factors should be taken into consideration during planning. Improvements in walking ability, basic activities of daily living, and gait and balance were noted after CGC irrespective of the presence of depressive symptoms. While balance and gait abilities in patients with depressive symptoms were moderate prior to CGC, patients eventually achieved the same functional level after the treatment as those without depressive symptoms. Female sex and chronic obstructive pulmonary disease were associated with the presence of depressive symptoms. However, further research is required to prove the relevance of depression in older adults undergoing CGC. In this context, a study including patients with and without depression equally distributed in both groups would be the best choice.

In addition to the current literature, our study provided important information on depression in elderly patients scheduled for CGC. It underlines the need for specific management of depression as part of CGC, requiring the involvement of specialized therapists and specific measures. For further research the specific treatment of depressive symptoms within the CGC should be in the focus with respect to the effectivity to combat depression and the related overall effect of CGC on short- and long-term outcomes.

## Figures and Tables

**Figure 1 geriatrics-08-00037-f001:**
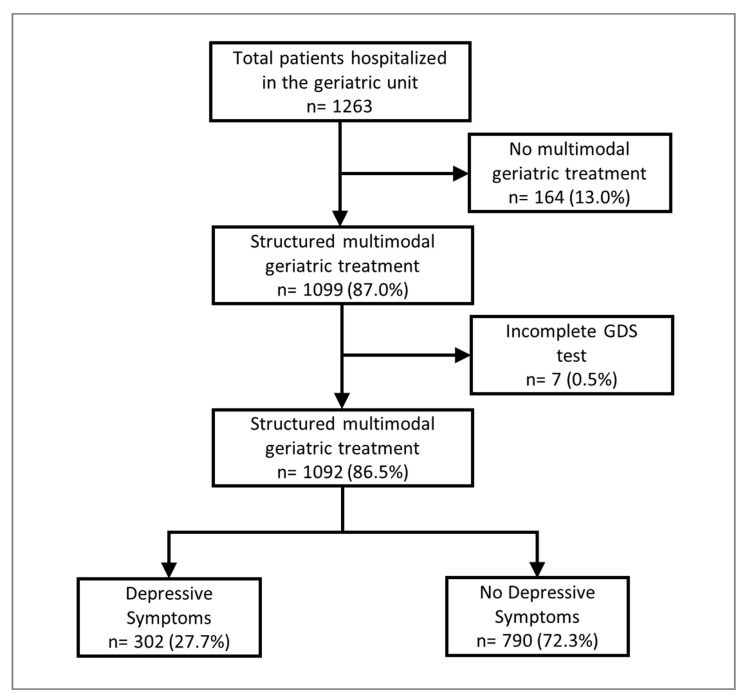
Patient selection. ‘GDS’ = Geriatric Depression Scale.

**Figure 2 geriatrics-08-00037-f002:**
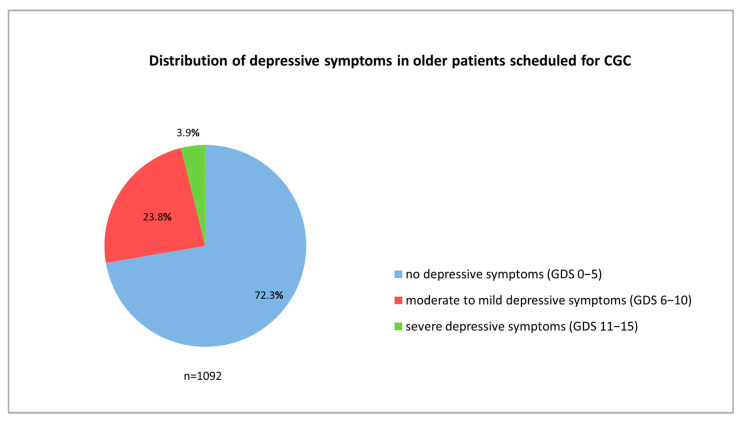
Distribution of depressive symptoms in older patients scheduled for comprehensive geriatric care. ‘GDS’ = Geriatric Depression Scale. ‘CGC’ = comprehensive geriatric care.

**Table 1 geriatrics-08-00037-t001:** Distribution of depressive symptoms among older adults treated with comprehensive geriatric care (*n* = 1092).

	Severe Depressive Symptoms	Mild Depressive Symptoms	No Depressive Symptoms
Patients (*n* = 1092)	*n* = 42 (3.9%)	*n* = 260 (23.8%)	*n* = 790 (72.3%)

**Table 2 geriatrics-08-00037-t002:** Timed Up and Go test, Tinetti score, and Barthel Index scores; values for geriatric patients with and without depressive symptoms prior to versus after comprehensive geriatric care (CGC).

	Prior to CGC	After CGC	*p **
**Patients with depressive symptoms**			
Timed Up and Go (median, IQR, years)	4 (3–5)	3 (3–4)	<0.001
Barthel Index (median, IQR, years)	45 (30–55)	55 (45–75)	<0.001
Tinetti score (median, IQR, years)	10 (4.75–14.25)	14 (8–19)	<0.001
**Patients without depressive symptoms**			
Timed up and go (median, IQR, years)	4 (3–5)	3 (2–4)	<0.001
Barthel Index (median, IQR, years)	45 (30–60)	55 (30–80)	<0.001
Tinetti score (median, IQR, years)	12 (6–17)	15 (2–20)	<0.001

‘CGC’ = comprehensive geriatric care; * *p*-value calculated in a Mann–Whitney U test.

**Table 3 geriatrics-08-00037-t003:** Elderly patients with and without depressive symptoms treated in a structured geriatric setting (comprehensive geriatric care).

	Total Group (*n* = 1092)	Depressive Symptoms (*n* = 302)	No Depressive Symptoms (*n* = 790)	*p **	*p ***
Age (median, IQR, years)	83.1 (79.1–87.7)	82.8 (79.0–87.3)	83.3 (79.0–88.0)	0.408	
Age ≥ 80 years	753 (69.0%)	212 (70.2%)	541 (68.5%)	0.608	
Sex					
Female	700 (64.1%)	210 (69.5%)	490 (62.0%)	0.024	0.046
Male	392 (35.9%)	92 (30.5%)	300 (38.0%)		
**Co-morbidities**					
Hypertension	849 (77.7%)	246 (81.5%)	603(76.3%)	0.074	
Diabetes mellitus	335 (30.7%)	99 (32.8%)	236 (29.9%)	0.379	
Heart insufficiency	258 (23.6%)	79 (26.2%)	179 (22.7%)	0.233	
Renal insufficiency	381 (34.9%)	118 (39.1%)	263 (33.3%)	0.076	
Coronary heart disease	279 (25.5%)	78 (25.8%)	201 (25.4%)	0.938	
Peripheral artery disease	59 (5.4%)	21 (7.0%)	38 (4.8%)	0.178	
Atrial fibrillation	387 (35.4%)	116 (38.4%)	271 (34.3%)	0.204	
Chronic obstructive pulmonary disease	108 (9.9%)	45 (14.9%)	63 (8.0%)	0.001	<0.001
Dementia	225 (20.6%)	48 (15.9%)	177 (22.4%)	0.019	0.062
Parkinson’s disease	62 (5.7%)	16 (5.3%)	46 (5.8%)	0.884	
Previous stroke	89 (8.2%)	27 (8.9%)	62 (7.8%)	0.539	
Current fracture	299 (27.4)	83 (27.5%)	216 (27.3%)	>0.999	
Osteoporosis	127 (11.6%)	39 (12.9%)	88 (11.1%)	0.401	
Vitamin B deficiency	484 (44.3%)	131 (43.4%)	353 (44.7%)	0.734	
**Short term adverse events while hospitalized**					
Diffuse pain	292 (26.7%)	85 (28.1%)	207 (26.2%)	0.541	
Delirium	57 (5.2%)	11 (3.6%)	46 (5.8%)	0.172	
Pneumonia	64 (5.9%)	23 (7.6%)	41 (5.2%)	0.149	
Urinary tract infection	158 (14.5%)	42 (13.9%)	116 (14.7%)	0.774	
Dizziness	79 (7.2%)	30 (9.9%)	49 (6.2%)	0.037	0.184
Deep vein thrombosis	5 (0.5%)	1 (0.3%)	4 (0.5%)	0.999	
Pulmonary embolism	4 (0.4%)	0 (0.0%)	4 (0.5%)	0.580	
Electrolyte imbalance	405 (37.1%)	120 (39.7%)	285 (36.1%)	0.264	
Hypokalemia	350 (32.1%)	108 (35.8%)	242 (30.6%)	0.111	
Hyponatremia	100 (9.2%)	27 (8.9%)	73 (9.2%)	0.999	
**Functional assessment on admission**					
Timed Up and Go (median, IQR; mean, SD) (*n* = 1063)	4 (3–5); 3.9 ± 1.1	4 (3–5); 4.1 ± 1.0	4 (3–5); 3.8 ± 1.1	0.001	
Barthel Index (median, IQR) (*n* = 1029)	45 (30–60)	45 (30–55)	45 (30–60)	0.392	
Tinetti Geriatric Assessment (median, IQR) (*n* = 994)	12 (5–17)	10 (4.75–14.25)	12 (6–17)	0.001	
MMSE (median, IQR) (*n* = 812)	26 (21–28)	26 (21–27)	26 (21–28)	0.290	
**Discharge type**					
In-home care	1035 (94.8%)	282 (93.4%)	753 (95.3%)	0.223	
Referral to other department	15 (1.4%)	5 (1.7.0%)	10 (1.3%)	0.537	
Length of hospital stay for patients with CGC (median, IQR, days) (*n* = 1029)	17 (16–19)	17 (16–19.25)	17 (16–19)	0. 147	

‘IQR’ = interquartile range; ‘SD’ = standard deviation; ‘CGC’ = comprehensive geriatric care. ** p*-value calculated in the univariate analysis. ** *p*-value calculated in the logistical regression analysis.

## Data Availability

For data availability and access, please contact: christian.tanislav@diakonie-sw.de.

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
