# Peer review of "Comprehensive Geriatric Care in Older Hospitalized Patients with Depressive Symptoms"

_geriatrics, 2023, doi:10.3390/geriatrics8020037_

Round 1

Reviewer 1 Report

Dear authors, Thanks for your efforts, please consider the following notes and suggestions,

On page 2 the authors stated that “In patients with myocardial

infarction, depression is a facilitator for health care consumption after the acute event”. Could you please explain more about how depression among patients with MI would facilitate healthcare consumption? And how comorbid MI with depression is related to the discussion of depression among older adults?

on page 2  the authors use the full phrase “ comprehensive geriatric care” twice and they already used the abbreviation “CGC” previously. Please keep using the abbreviation throughout the text for the matter of saving space and not confusing the reader.

Please consider using a conceptual framework that shows the relationships between CGC, depression, and health outcomes among older adults. This would be very helpful for you and the readers.

Could you please explain why you chose to use median and IQR to describe continuous variables instead of means and standard deviations in your study?

I guess that the percentage in this statement was miscalculated “ and severe

depressive symptoms (GDS 11–15) in 42 patients (23.8%)” I think it should be (3.9%) of the total sample size.

I suggest that one of the future recommendations for similar studies is select a purposive sample of equally distributed older adults with and without depression. 

Author Response

See uploaded file

Reviewer 2 Report

The objective in the abstract is different from the objective described at the end of the introduction. According to reading the article, the objective described at the end of the introduction is more consistent with the study.

It remains to describe the type of study.

The author must describe  why he performed logistic regression and linear regression in the methodology.

Author Response

see uploaded file

Reviewer 3 Report

Comprehensive geriatric care in older hospitalized patients with depressive symptoms

Title

-       Adequate

Abstract

-       No connection with the previous idea “the specific multimodal treatment for older patients”

-       Do not put acronyms in the abstract

-       In the method, the authors should better specify the type of study, place of study, instruments used, sample, analysis performed.

-       Improve the description of the results

-       The conclusions do not respond to the objective of the study

-       Does not show descriptors

Introduction

-       The authors should better describe the definition of depression used and its pathophysiology and prevalence. They bring examples of how depression does not contribute to the improvement of the elderly.

-       In addition, they must describe or define what CGC is.

-       It is suggested to improve the justification of the study, why it is important for the health team to work with the CGC

-       Improve the objective of the study, the very fact of carrying out the study is to investigate and not a verb for the general objective.

Methods

-       What type of study was carried out? Better describe the population with inclusion and exclusion criteria.

-       Describe the place of study

-       Missing description of the MMSE as an assessment tool

-       Improve the description of the statistical analyzes used in the study

Results

-       Create a table with participant information. It is suggested to remove the pie chart and only keep the prevalence of depressive symptoms and their categories found in the study as described.

-       Information that does not appear in the method was identified, such as the type of morbidities collected.

-       Why the authors did not perform multiple logistic regression, knowing that it has categorical and numerical information. Has the regression been fitted? since there are confounding variables such as age and sex.

-       How would the interpretation of the information in Table 3?

-       It is not clear how the information was analyzed, does the study state that there were two collections? if the study is longitudinal, were the analyzes performed with the difference between the second and first evaluation?

Discussion

-       The first paragraph repeats the results; it is recommended to rewrite it with the most important information from the study.

-       Is it unclear how CGC decreases depressive symptoms in the elderly? Was there any kind of intervention? As they do not explain what CGC is, the results are not clear.

-       Digital experiences appear, but this was not observed in the results. Focus on findings.

-       The authors compare with other studies, but do not deepen the discussion, why did they find these findings in this population?

-       Do they discuss data that were not significant, table 2, fractures, and what are the main results in Table 3? Review the discussion.

-       Review study limitations, was a control group necessary? Was that the biggest limitation?

-       The contribution of CGC was not identified in the study, as indicated by the authors,

Conclusion

-       Partially responds to the objective of the study,

-       What is the contribution of the study to the advancement of the area?

References

adequates 

Author Response

see uploaded file
